# Evaluation of Garlic Juice Processing Waste Supplementation in Juvenile Black Rockfish (*Sebastes schlegelii*) Diets on Growth Performance, Antioxidant and Digestive Enzyme Activity, Growth- and Antioxidant-Related Gene Expression, and Disease Resistance against *Streptococcus iniae*

**DOI:** 10.3390/ani12243512

**Published:** 2022-12-12

**Authors:** Hwa Yong Oh, Tae Hoon Lee, Da-Yeon Lee, Chang-Hwan Lee, Min-Young Sohn, Ryeong-Won Kwon, Jeong-Gyun Kim, Hee Sung Kim, Kyoung-Duck Kim

**Affiliations:** 1Department of Marine Biology and Aquaculture, Gyeongsang National University, Tongyeong 53064, Republic of Korea; 2Department of Seafood Science and Technology, Gyeongsang National University, Tongyeong 53064, Republic of Korea; 3Southeast Sea Fisheries Research Institute, National Institute of Fisheries Science, Tongyeong 53017, Republic of Korea

**Keywords:** garlic juice processing waste (GJPW), growth performance, digestive enzyme, antioxidant response, disease resistance, black rockfish (*Sebastes schlegelii*)

## Abstract

**Simple Summary:**

Garlic juice is one of the most popular health drinks in Asia. Garlic juice processing waste (GJPW) is a potential feed ingredient for aquaculture, because it is rich in bioactive chemicals, which makes it an attractive alternative to synthetic antibiotics and/or antioxidants. This study investigates the effects of various dietary levels of garlic juice processing waste on the growth performance and health status of juvenile black (*Sebastes schlegelii*). On the basis of the results, rockfish fed diets supplemented with GJPW show improvement in growth performance, digestive enzyme activity, and growth and antioxidant-related gene expression. Therefore, considering the effects on the overall performance of juvenile rockfish, GJPW is a promising useful additive, and 2.5 g kg^−1^ dietary GJPW was found to be a suitable dietary level for juvenile rockfish black rockfish.

**Abstract:**

An 8-week feeding trial was conducted to evaluate the effects of various dietary levels of garlic juice processing waste (GJPW) on the growth, feed utilization, digestive and antioxidant enzyme activity, growth- and antioxidant-related gene expression, and resistance to *Streptococcus iniae* infection of juvenile black rockfish (*Sebastes schlegelii*). A total of 450 juvenile rockfish were randomly distributed into 30 L rectangular tanks (30 fish per tank). Five experimental diets were prepared in triplicate. The fish were fed experimental diets supplemented with GJPW at concentrations of 0 (GJPW0, control), 2.5 (GJPW2.5), 5 (GJPW5), 7.5 (GJPW7.5), and 10 g kg^−1^ (GJPW10) diet. All of the GJPW-supplemented treatments (2.5, 5, 7.5, and 10 g kg^−1^) significantly enhanced weight gain (WG), specific growth rate (SGR), feed efficiency (FE), protein efficiency ratio (PER), and digestive enzyme activity (amylase, trypsin, and lipase). A decreasing trend was seen in plasma aspartate aminotransferase (ALT), alanine aminotransferase (AST), and glucose (GLU) content with increasing dietary levels of GJPW. In contrast, plasma lysozyme and antioxidant enzyme activities were significantly increased with increasing dietary GJPW levels. Furthermore, GJPW administration significantly upregulated the expression of insulin-like growth factor-1 (IGF-1), superoxide dismutase (SOD), catalase (CAT), and glutathione S-transferase (GST) in the liver of rockfish. A challenge test with *S. iniae* showed significantly higher resistance in the GJPW-supplemented treatments than in the control. In short, dietary supplementation GJPW enhanced growth performance and antioxidant response in juvenile black rockfish, with suitable effects in fish fed with 2.5 g kg^−1^ GJPW for 8 weeks.

## 1. Introduction

Aquaculture, which will have a population of 9.7 billion by 2050, is unquestionably the fastest growing animal protein source in the world [1]. Intensification of aquaculture has led to stressful conditions and has been the consequence of the outbreak of disease [2,3]. While synthetic antibiotics have been widely employed to resolve this concern for years, antibiotics in aquaculture can have a negative influence on the environment and contribute to the development of bacterial resistance to numerous drugs [4]. As a result, several recent research studies on the use of plant-based products as a feed additive in the aqua-feed industry have concentrated on improving growth performance, digestive system function, immunological parameters, and bacterial resistance [5,6,7]. Due to their active biomolecules such as alkaloids, phenolic compounds, sterols, flavonoids, glycosides, essential oils, saponins, and terpenes, phyto-additives enhance the health and growth of fish and also provide antioxidant activity [8,9]. Garlic (*Allium sativum* L.) has been used for culinary and medicinal purposes throughout history [10,11]. Garlic components contain a diverse array of bioactive compounds, including allicin (diallyl thiosulfate), which imparts garlic’s characteristic pungent aroma and medicinal properties [12], as well as vitamins (ascorbic acid, thiamine and riboflavin), minerals (potassium, phosphorus, calcium, magnesium, sodium, iron, selenium, and germanium), flavonoids (phenolic acids) [13], amino acids [14], steroidal saponins [15] and phytosterols [15], which present antibacterial, antiviral, antiparasitic, immunostimulatory and antioxidant properties [16,17,18]. Previous studies have shown that dietary supplementation of garlic contributed to growth performance [19,20], immune responses [21,22,23], antioxidant status [24] and disease resistance [16,25,26] in fish.

Garlic juice is one of the most popular health drinks in Asia [27]. In Korea, garlic juice is commonly commercially available for more than 2,000 won (KRW, $1 USD) per 100 mL, and there are many garlic juice producers [28]. However, garlic juice processing waste (GJPW) are considered garbage or used as fertilizer [29]. In addition to contributing to the sustainable development of the aquaculture industry due to the use of waste from garlic juice production that can be discarded from the environment, contaminated soil, and water course, it can be a viable alternative in a practical fish formulated diet [30].

Black rockfish (*Sebastes schlegelii*) is one of the most important mariculture fish in the Wester North Pacific due to its high economic and ecological values [31,32]. In particular, its annual aquaculture production in Korea was 17,473 tons, the second largest production among the marine finfish aquaculture in 2021 [33]. However, various diseases, such as lymphocystis disease virus, *Streptococcus iniae*, *Vibrio anguillarum*, *Edwardsiella tarda*, and *Aeromonas salmonicida*, etc., have threatened sustainable production, causing enormous economic losses to the culture industry of *S. schlegelii* [34,35,36].

In a recent study, growth performance, non-specific immunity, and resistance against *V. harveyi* infection in juvenile *S. schlegelii* were notably improved by dietary 1% addition of GJPW [37]. Although applicability of GJPW as the functional feed additive in *S. schlegelii* feed was successfully reported [37], their suitable inclusion levels in formulated diet on rockfish must be proved before a practical application. To the best of our knowledge, no dietary information is available about the different GJPW levels on growth performance and health status of *S. schlegelii*. Thus, the aim of this study was to obtain a GJPW that could be used as a source of feed additives for black rockfish juveniles and its graded level of effects on growth, feed utilization, antioxidant and digestive enzyme activities, growth- and antioxidant-related gene expression, and disease resistance against *S. iniae*.

## 2. Materials and Methods

### 2.1. Preparation of Experimental Diet

GJPW were provided from Youngjin Healthy Juice Store (Daegu, Korea). GJPW were dried for 72 h at 20 °C in an agricultural product dryer (KED-M07D1, Kiturami Co., Ltd., Seoul, Korea) for 72 h, then crushed into a fine powder, and kept at −20 °C. After processing, the composition of GJPW were analyzed in laboratory and the result indicated that the GJPW retained certain functional compounds, such as total phenolic (27.3 mg gallic acid 100 g sample^−1^), flavonoids (36.8 mg 100 g sample^−1^). Additionally, GJPW clearly displays a dose-dependent DPPH and ABTS radical scavenging activities (Table 1).

Five isonitrogenous and isolipidic experimental diets were formulated with the supplementation of 0 g kg^−1^ diet (GJPW0), 2.5 g kg^−1^ diet (GJPW2.5), 5 g kg^−1^ diet (GJPW5), 7.5 g kg^−1^ diet (GJPW7.5) and 10 g kg^−1^ diet (GJPW10) GJPW (Table 2). The main protein sources were pollock meal and fermented soybean meal. The fish and soybean oils, and wheat flour provided as sources of lipid and carbohydrate, respectively. Dry ingredients of all experimental diets were mechanically mixed to insure homogeneity. Then, to form dough, fish and soybean oils and distilled water were added to the mixture. The moist dough was then chopped (3.0 mm diameter, SL Machinery, Incheon, Korea) to prepare pellets. The pellets were dried at 20 °C for 48 h in an agricultural product dryer (KED-M07D1, Kiturami Co., Ltd., Seoul, Korea) and then stored at −20 °C in a freezer until use.

### 2.2. Feeding Trial

Black rockfish were acquired from a commercial hatchery (Namhae-gun, Gyeongsangnam-do, Korea) and taken to the Marine Bio-Education and Research Center (Gyeongsang National University, Tongyeong-si, Gyeongsangnam-do, Korea). Fish were kept in the experimental conditions for two weeks and fed a commercial diet (Jeil Feed Co., Haman-gun, Gyeongsangnam-do, Korea; 52% crude protein and 10% crude lipids) two times a day before feeding trial. There were 450 juvenile rockfish with an average body weight of 2.2 g that were starved for 24 h before being randomly distributed in 15 flow-through rectangular tanks (30 L), each tank held 30 fish. For eight weeks, three replicate groups of fish were fed the five experimental diets twice daily at 09:00 and 17:00. During the feeding trial, the mean water temperature, dissolved oxygen and salinity were 21.2 ± 0.22 °C, 7.0 ± 0.06 mg/L and 30.13 ± 0.12 psu, respectively. The photoperiod was set to follow natural conditions, each tank had constant aeration.

### 2.3. Sample Collection

At the end of the feeding trial, all fish were deprived of feed for a day and were anesthetized with tricaine methane sulphonate (MS-222) at 200 ppm before sampling and counted and individual weight and length were recorded to measure growth parameters. Five fish were randomly taken from each tank (15 fish per group) for chemical composition analysis. Blood samples were obtained from the caudal veins of ten anesthetized fish in each tank using heparinized syringes and gently transferred to 1.5 mL Eppendorf tubes. Plasma was collected by centrifugation for 10 min at 7,500 rpm. Plasma samples were stored at −80 °C in a freezer until analysis. Fish were dissected immediately after blood sampling to collect the liver and viscera for calculating condition parameters. Then, intestine and liver were rinsed with cold distilled water and stored at −80 °C until further digestive enzyme and gene expression analysis, respectively.

### 2.4. Challenge Test

After sampling, ten fish were randomly selected for the challenge test from each tank. The *S. iniae* strain was provided by the Korean Culture Collection of Aquatic Microorganisms, National Institute of Fisheries Science (Busan, Korea). All fish were artificially injected via intraperitoneal injection with 0.1 mL of pathogenic *S. iniae* culture suspension at a concentration of 5.0 × 10^6^ CFU/mL. The injection concentration was based on our preliminary results (LD50 4.96 × 10^6^ CFU/mL). The water temperature was maintained at 20.5 ± 0.02 °C (mean ± SE) with dissolved oxygen (7.1 ± 0.03 mg/L) and, fish survival was monitored daily for six days throughout the challenge test. Dead fish were removed every 12 h throughout the observation period.

### 2.5. Analysis

#### 2.5.1. Chemical Analysis

The total phenolic content of GJPW was determined using a Folin-Ciocalteu regent [38]. The 50 μL of the sample was vortexed with 1 mL Folin-Ciocalteu reagent. After 3 min reaction time, the 1 mL of 10% sodium carbonate solution was added to the mixture. After 60 min of incubation at room temperature, the absorbance at 700 nm was determined. The standard was gallic acid and quercetin (Sigma-Aldrich Co., St Louis, MO, USA).

The flavonoids content of GJPW was determined following the method described by Moreno et al. [39]. The 1 mL of sample was diluted with 0.1 mL of 10% aluminum nitrate, 0.1 mL of 1 mol/L aqueous potassium acetate, and 4.3 mL of 80% ethanol. The absorbance at 415 nm was determined After 40 min at room temperature in a dark room. The quercetin (Sigma-Aldrich Co., St Louis, MO, USA) was used as the standard.

The DPPH scavenging activity was assessed using the method of Blois [40]. To summarize, 80 μL of sample or ascorbic acid as positive control was mixed with 100 μL of 150 μM DPPH in methanol, and the mixture was kept in the room temperature for 10 min. The absorbance at 525 nm was evaluated using a microplate reader (SpectraMax^®^ M2/M2e, Sunnycale, CA, USA).

The radical scavenging activity of ABTS was determined using the method reported by Re et al. [41]. Briefly, ABTS radical was created by incubating 7 mM ABTS with 2.4 mM potassium persulfate in the dark for 16 h, and then the working solution was prepared by diluting with distilled water to absorbance of 1.5 at 414 nm. A 50 μL of sample or ascorbic acid as positive control and 100 μL of the working solution were mixed, allowed to stand for 5 min at the room temperature, and the absorbance at 414 nm was measured.

The proximate composition of the experimental diet and whole-body samples were determined using the procedures recommended by the Association of Official Agricultural Chemists [42]. The crude protein and crude lipid contents were determined by the Kjeldahl method and Soxhlet extraction methods using a KD310–A–1015 KjelROC Analyzer (OPSIS Liquid LINE, Skytteskogsvägen, Furulund, Sweden) and Soxtec extractor (ST 243 Soxtec™; FOSS, Hillerod, Denmark), respectively. Moisture content was determined by oven drying at 105 °C for 24 h, and ash was determined using a muffle furnace at 600 °C for 4 h.

#### 2.5.2. Plasma Chemistry Analysis

The activity of aspartate aminotransferase activity (AST) and alanine aminotransferase activity (ALT), total cholesterol (T-CHO), total protein (TP) and glucose (GLU) were determined using an automatic chemistry system (Fuji Dri-Chem NX500i; Fujifilm, Tokyo, Japan).

#### 2.5.3. Digestive Enzymes Measurements

Stored intestine tissue were homogenized in 10 volumes (*v*/*w*) of ice-cold, 0.86% physiological saline in an ice bath with a TissueLyser II (QIAGEN, Venlo, Netherlands), then centrifuged at 13,000 rpm for 10 min at 4 °C to obtain the supernatant. Amylase, trypsin and lipase activities were measured using a commercial kit (Abcam, Trumpington, Cambridge, UK). All digestive enzymes activity were determined according to the manufacturer guidelines with a spectrophotometer (Thermo Scientific MULTISKAN GO, Vantaa, Finland).

#### 2.5.4. Lysozyme and Antioxidant Enzyme Activities Analysis

The activity levels of plasma lysozyme were measured using a commercial kit (EnzChek™ Lysozyme Assay Kit, Thermo Fisher Scientific, Waltham, MA, USA) in accordance with Galagarza et al. [43]. Fluorescence intensity was determined with a fluorescence reader (1420 Multilabel Counter Victor3, Perkin Elmer, Waltham, MA, USA) at excitation/emission wavelengths of 485/535 nm.

Superoxide dismutase (SOD) activity was measured using a Cayman’s Superoxide Dismutase Assay Kit (Cayman Chemical, Ann Arbor, MI, USA) following the manufacturer’s instructions. Briefly, 10 μL of plasma was added to 200 μL of the radical detector. The reaction was initiated by adding 20 μL of xanthine oxidase, and the resultant mixture was incubated at room temperature on a shaker for 20 min. The absorbance was measured at 440 nm using a spectrophotometer (Thermo Scientific MULTISKAN GO, Vantaa, Finland). Catalase (CAT) activity was analyzed using a Cayman’s Catalase Assay Kit (Cayman Chemical, Ann Arbor, MI, USA) following the manufacturer’s instructions. Briefly, 20 μL of plasma was added to 30 μL of methanol and 100 μL of assay buffer. The reaction was initiated by adding 20 μL of H_2_O_2_, and the mixture was incubated at room temperature for 20 min. To terminate the reaction, 30 μL of potassium hydroxide and purpald chromagen were added, and the mixture was incubated at room temperature for 10 min. Finally, 10 μL of potassium periodate was added and incubated at room temperature on a shaker for 5 min. The solution absorbance was measured at 540 nm (Thermo Scientific MULTISKAN GO, Vantaa, Finland). Glutathione (GSH) concentration was measured using a Cayman’s GSH Assay Kit (Cayman Chemical, Ann Arbor, MI, USA) following the manufacturer’s instructions. Briefly, 150 μL of freshly prepared assay cocktail containing MES buffer [2-(N-morpholino)ethanesulfonic acid], cofactor mixture, enzyme mixture, water, and DTNB [5,5′-dithio-bis-(2-nitrobenzoic acid)] was added to 50 μL plasma samples in a 96-well plate. The absorbance was measured at 405 nm at 5 min intervals for 30 min (Thermo Scientific MULTISKAN GO, Vantaa, Finland).

### 2.6. Expression of Growth and Antioxidant-Related Genes

#### 2.6.1. Primer Design

For quantitative PCR (qPCR) assay the specific primer pairs were designed using the NCBI Genbank to investigate the effects of GJPW on the expression of insulin-like growth factor (IGF-1), SOD, glutathione S-transferase (GST) and CAT (Table 3). In this study, β-actin gene was used as a housekeeping gene to normalize the expression levels of the selected genes in black rockfish.

#### 2.6.2. Total RNA Extraction and qPCR Assay

The stored liver was used to isolate the total RNA according to the RNAiso Plus (TaKaRa, Kusatsu, Shiga, Japan) manufacturer’s recommendations. Subsequently, genomic DNA contamination was removed by treatment with recombinant DNase I (TaKaRa Bio Inc., Kusatsu, Shiga, Japan), and the quantity and concentration of purified total RNA were measured by a NanoVue (GE Healthcare, Chicago, IL, USA) spectrophotometer. The purified samples were synthesized with cDNA following the protocol using PrimeScript 1st strand cDNA Synthesis Kit (TaKaRa Bio Inc., Japan).

The expression level of IGF-1, SOD, GST, and CAT was measured using specific primer, B Green Premix Ex Taq™ (TaKaRa Bio Inc., Japan) and Thermal Cycler Dice Real Time System III (TaKaRa Bio Inc., Japan). A melting curve was run at the end of the 45 amplification cycles to test for the presence of a unique PCR product. All qPCR data were measured and presented relative to the cycle threshold (Ct) values and then converted to fold changes by 2^−∆∆Ct^ method [44]. The iQ5 optical system (BioRad, Hercules, CA, USA) was used for the data analysis.

### 2.7. Calculations and Statistical Analyses

To calculate fish performance parameters using the following formula:Survival (SR, %) = (number of fish at the end of the trial/number of fish at the beginning of the trial) × 100
Weight gain (WG, g/fish) = final body weight − initial body weight
Specific growth rate (SGR, %/day) = [ln final weight of fish − ln initial weight of fish)/days of feeding] × 100
Feed consumption (FC, g/fish) = total dry feed intake/number of surviving fish
Feed efficiency (FE) = WG of fish/feed consumed
Protein efficiency ratio (PER) = WG of fish/protein consumed
Protein retention (PR, %) = Protein gain/protein consumed × 100
Condition factor (CF) = Fish weight/total length^3^ × 100
Hepatosomatic index (HSI, %) = (liver weight/whole-body weight) × 100
Viscerosomatic index (VSI, %) = (viscera weight/whole-body weight) × 100

All percentage values were arcsine-transformed before analysis. The results were expressed as the mean and pooled standard error of the means. The Levene’s test was used to test the homogeneity of variances among treatments. After that, the data were subjected to one-way analysis of variance (ANOVA) by Duncan’s multiple range test [45] were analyzed at significant differences (*p* < 0.05). Also, orthogonal polynomial contrasts (linear, quadratic, and cubic) were used to evaluate the response for all dependent variables [46], and the data were subjected to regression analysis to fit the best model when the statistical significance was detected. Fish survival during the 6-days post-observation period after artificial *S. iniae* injection was analyzed using Kaplan–Meier survival curve, Log-rank and Wilcoxon tests. All statistical analyses were carried out using SPSS version 25.0 program statistical software package (SPSS Inc., Chicago, IL, USA).

## 3. Results

### 3.1. Growth, Feed Utilization, and Biological Parameters

After 8-weeks of the feeding trial, growth performance and biological parameters of rockfish fed the experimental diets were shown in Table 4.

The WG and SGR of rockfish exhibited an increase with increasing GJPW content in a linear model, both of which were shown to be significantly higher in the GJPW2.5, GJPW5, GJPW7.5 and GJPW10 treatments than in the GJPW0 treatment (*p* < 0.05). However, graded levels of dietary GJPW presented no significant effects on SR, CF, VSI and HSI of juvenile rockfish. With the gradual increase of dietary GJPW level, the FE showed a linear trend (*p* < 0.05). The FE of fish fed the GJPW0 was significantly lower than that of fish fed the GJPW2.5, GJPW5, GJPW7.5 and GJPW10 treatments. As dietary GJPW level increased gradually, the PER and PR significantly increased linearly (*p* < 0.05). The PR was significantly higher in the GJPW supplemented treatments than that in the control (GJPW0) treatment (*p* < 0.05). However, the PR of fish fed GJPW7.5 and GJPW10 was significantly higher than that in GJPW0, but did not significantly differ from that in GJPW2.5, GJPW5 and GJPW7.5 treatments. No relationship was found for FC among treatments.

### 3.2. Whole-Body Proximate Composition

As shown in Table 5, the whole-body moisture, crude protein, crude lipid, and ash contents of fish fed the experimental diet for 8-weeks showed no significant difference among treatments.

### 3.3. Hematological Parameters

As dietary GJPW level increased gradually, the AST, ALT, and GLU contents showed a linearly decreasing trend (*p* < 0.05), while T-CHO and TP showed no significant difference among groups (Table 6). The AST, ALT, and GLU were significantly lower in the GJPW2.5, GJPW5, GJPW7.5 and GJPW10 treatments than in the GJPW0 treatment (*p* < 0.05). The activities of lysozyme, SOD and CAT, and GSH content in plasma in fish all showed significantly increasing linear model in response to the dietary GJPW level (*p* < 0.05). All treatment groups showed a higher lysozyme, SOD and CAT activity, and GSH content than that of the control treatment group with a significant difference (*p* < 0.05).

### 3.4. Digestive Enzyme Activities

Amylase, trypsin, and lipase activities of fish fed the experimental diet for 8 weeks were listed in Table 7. The activities of amylase, trypsin, and lipase enzyme in the intestine of rockfish all showed a positive linear trend in response to dietary GJPW level (*p* < 0.05). The amylase, lipase and trypsin enzyme activity were significant improvement among all fish fed with GJPW supplemented diets compared to the control group (*p* < 0.05).

### 3.5. Genes Expression by Quantitative Real Time PCR

The mRNA levels of the growth (IGF-1) and antioxidant (SOD, GST, and CAT) genes in each treatment in comparison with the GJPW0 treatment were shown in Figure 1. IGF-1 mRNA transcription level of fish fed the GJPW2.5, GJPW5, GJPW7.5, and GJPW10 diets significantly increased compared to GJPW0 diets (*p* < 0.05). The highest expression level of IGF-1 gene was shown in the fish fed with GJPW10 diet compared to other treatments. SOD, GST, and CAT gene expressions were significantly upregulated in the GJPW2.5, GJPW5, GJPW7.5 and GJPW10 treatments compared to the GJPW0 treatment (*p* < 0.05).

### 3.6. Challenge Test

The Figure 2 shown that survival of fish artificially infected with *S. iniae* was observed for 6 days post-infection. Survival of fish fed the GJPW0 diet was significantly (*p* < 0.05) lower than that of fish fed the GJPW2.5, GJPW5, GJPW7.5, and GJPW10 diets.

## 4. Discussion

After the process of manufacturing healthy juice of garlic, known as medicinal plants, wastes also contained a large number of functional compounds derived from many studies, such as phenols and flavonoids [18,47,48,49]. Phenolic compounds and flavonoids are widely known for their beneficial effects on the growth, feed utilization, antioxidant activities, immunity, and disease resistance of fish, thus enhancing their health [50,51,52]. In this study, GJPW used as feed additives also showed sufficient phenols and flavonoids content as well as antioxidant activities even after the juice process of the raw materials. These bioactive compounds can attribute to improve the health of fish, as found in this study for juvenile rockfish fed the dietary supplements with GJPW. In addition to its bioactive compounds, garlic is rich in organosulfur compounds [53,54]. The most common organosulfur compounds include S-allyl cysteine, S-allyl-mercaptocysteine, and allicin, which were abundant in garlic extracts, maceration, and juice [55,56]. Garlic also has other bioactive compounds such as polysaccharides and fructans [57,58]. In fish culture, these bioactive compounds present antibacterial, antiparasitic, antioxidant, immunostimulatory, and growth promoting activities [17]. The presence of these compounds and antioxidant activities (DPPH and ABTS) in GJPW may have contributed to the performance of juvenile black rockfish. 

In this study, the growth (WG and SGR) parameters and feed utilization (FE, PER and PR) indices at 2.5–10 g kg^−1^ in treated fish with dietary GJPW improved after 8-weeks feeding trial. These results means that the GJPW supplementation improved feed utilization, thus increasing WG and SGR. These results were similar to other studies on the addition of garlic in fish feed. Aly and Atti [59] found that tilapia (*Oreochromis niloticus*) fed with a diet supplemented with 10 and 20 g kg^−1^ of the crushed garlic diet for 8-weeks showed increased growth by improving feed utilization. Thanikachalam et al. [60] reported that catfish (*Clarias gariepinus*) fed diets containing garlic husk powder of 5, 10 and 15 g kg^−1^ diet for 20 days showed a significant effect in growth and feed utilization compared to the control diet. Especially, the supplementation of garlic in fish feed can improves growth performance due to allicin, which is a potent stimulant for the chemoreception, which increases the feed intake and/or feed utilization in fish [17]. In addition, the active components, such as phenolic compounds from plant additive could improve the digestibility and the availability of nutrients resulting in an increase in feed utilization [61,62,63]. Furthermore, this study showed that the alteration in digestive enzyme activities in intestine, which significantly increased in fish fed the supplemented GJPW diet, resulting in improved utilization of nutrients. Hassaan et al. [64] showed that inducing the secretion of intestinal digestive enzymes by the fish may increase the digestion and absorption of the nutrients of feed. This evidence indicated that the use of GJPW in feed has a positive effect on growth performance by maximizing feed utilization by activating the action of digestive enzymes in fish.

Additionally, there was high correlation between IGF-1 gene expression and growth performance in this study. In general, IGF-1 is mainly secreted from the liver after releasing the growth hormone [65]. In this vein, nutritional status can affect the growth hormone-IGF axis, and this may influence the acceleration of somatic growth in fish [2,66,67]. The higher growth performance was recorded by fish fed diet supplemented with GJPW compared to the control (GJPW0) treatment, and the relative expression of IGF-1 gene mRNA exhibited the same trend. This finding indicated that the supplementation of 2.5 to 10 g of GJPW in diet can have a positive regulatory effect on the transcription of the IGF-1, which can eventually involve hyperplastic and hypertrophic muscular growth. Our result was similar to that of Mostafavi et al. [1], Ramezani et al. [68] and Safari et al. [69] who reported that the relative expression of growth-related gene was significantly up-regulated in fish fed diet supplemented with plant additives. In addition, Fuentes et al. [70] noted that fish have been made transgenic for the growth-related gene showed significantly improved growth by muscle hypertrophy and hyperplasia. Moreover, in this study, GJPW enhanced growth factors by feed utilization and stimulated growth-related gene expression. However, further study on the dietary GJPW supplementation effect on the physiological properties of fish is needed in the future.

Our results revealed that dietary GJPW supplementation did not significantly affect the whole-body composition of fish. Previous studies reported that the whole-body composition could reflect fish quality and is influenced by several factors including feed composition and feeding strategy [71]. Several studies recorded significant influence in the whole-body composition after being fed a garlic-supplemented diet such as feeding trials in in rainbow trout, *Oncorhynchus mykiss* [19,23]; African catfish, *C. gariepinus* [72]; red belly tilapia, *Tillapia zillii* [73]; Nile tilapia, *O. niloticus* [59]; Asian sea bass, *Lates calcarifer* (Bloch) [74]; and starlet sturgeon, *Acipenser ruthenus* [75]. The reason for this discrepancy in result can be attributed to different fish species and culture conditions or different ingredients of diets.

The hematologic parameters indicated the health and nutritional status in fish [76,77]. Some studies showed that hematological parameters were not always altered by fish diet [78]. In this study, hematological parameters (AST, ALT and GLU) showed significant differences among the hematologic values of fish fed with or without GJPW. Generally, AST exists in hepatocyte mitochondria, while ALT is spread around the hepatic cells and bile duct [79]. Increased serum AST and ALT activity in fish indicates an enzyme leak in the damaged plasma membrane and/or an increase in enzyme synthesis by liver tissue [80]. Therefore, the activity of AST and ALT is used as important indicators of liver health and their functions [81]. In addition, it can also be used to asses fish health status and stress indicators [82]. In this study, plasma ALT and AST activities decreased with increasing dietary level of GJPW, which indicated that a better tendency occurred in the hepatocytes of rockfish. In addition, the lowest levels of AST and ALT may be associated with liver protection properties in which antioxidants such as flavonoids. Phenols inhibit the release of liver damage enzymes into plasma through antioxidant activity and lipid peroxidation prevention of cell membranes [83]. In this context, natural antioxidant flavonoids improved the liver function and enhanced its ability against oxidative stress and tissue damage [84,85,86].

Generally, an increased immune response is affected by nutrients in fish diet [87]. In line with this study, the increase in plasma lysozyme activity was observed with the addition of 2.5 to 10.0 g GJPW per kg diet. In fish, the non-specific immune system was improved by increasing plasma lysozyme activity [6,78]. In particular, previous studies presented that garlic-added feed enhanced lysozyme, then improved immune capability [21,74]. Moreover, fish antioxidant systems were made of non-enzymatic compounds and antioxidant enzymes including SOD and CAT [1,88]. In this study, fish fed the addition of 2.5 g GJPW per kg diet showed a significant improvement in plasma SOD and CAT values, as well as fish fed supplements with GJPW higher than GJPW0 diet (control diet). The phenols and saponin contained in garlic were known to provide antioxidant activity [18,47,48]. Previous study has also demonstrated that plant flavonoids, such as ferns [88], orange peel [89], and *A. mongolicum Regel* [90] flavonoids, have strong antioxidant activity. These compounds can inhibit the formation of free radicals, enhance the absorption mechanism of endogenous radicals, and increase cellular antioxidant enzymes, such as SOD, CAT and glutathione peroxidase, and hemeoxygenase-1 [90]. Similar to this study, Nile tilapia (*O. niloticus*) fed with diets containing different garlic sources; natural garlic (40 g kg^−1^ diet), capsule of garlic oil (250 mg kg^−1^ diet), and garlic powder (32 g kg^−1^ diet) revealed improved activities of SOD and CAT [91]. Also, dietary garlic powder could significantly improve SOD and CAT activities in common carp (*C. carpio*) [91]. Furthermore, the expression of genes (SOD, GSH and CAT) related to the antioxidant defense system were examined to investigate further the molecular mechanisms of GJPW improved antioxidant responses in this study. Results showed that all tested genes encoding antioxidant enzymes in liver were associated with significant upregulation of fish fed with GJPW. The upregulation of antioxidant related genes may contribute to an increase in their enzyme activities [92].

Currently, bacterial challenge tests were often used as a final indicator of fish health status after nutrition trials [93], since the methodology to comprehensively investigate immunity and disease resistance of fish is still limited, and an effective biomarker for disease resistance of fish has been difficult to identify [94]. In this study, dietary 2.5 to 10 g kg^−1^ GJPW had a significantly enhanced survival rate of rockfish compared to the challenge with *S. iniae*. is a Gram-positive bacterium that is known to cause serious Streptococcosis to aquaculture worldwide [95]. Previous studies reported that the various types of garlic additives can enhance resistance of fish to bacterial pathogens [23,60,74]. Particularly, Lee et al. [37] presented that a diet supplemented with 1% garlic juice extraction waste improved the survival rate of *S. schlegelii* against Gram-negative *V. harveyi*. This promotion of disease resistance against Gram-negative and Gram-positive bacteria of juvenile rockfish contributed to the administration of bioactive compounds in GJPW and to the regulation of lysozyme activity and antioxidant capacity.

## 5. Conclusions

This study indicates that juvenile rockfish fed diets supplemented with GJPW show improvement in growth performance and hemato-physiological variables. In the meantime, the digestive enzyme activity, growth, and antioxidant-related gene expression were also increased by the supplementation of feed with GJPW. In conclusion, considering the effects on the overall performance of juvenile rockfish, GJPW is a promising function additive and 2.5 g kg^−1^ dietary GJPW was found to be a suitable dietary level for black rockfish.

## Figures and Tables

**Figure 1 animals-12-03512-f001:**
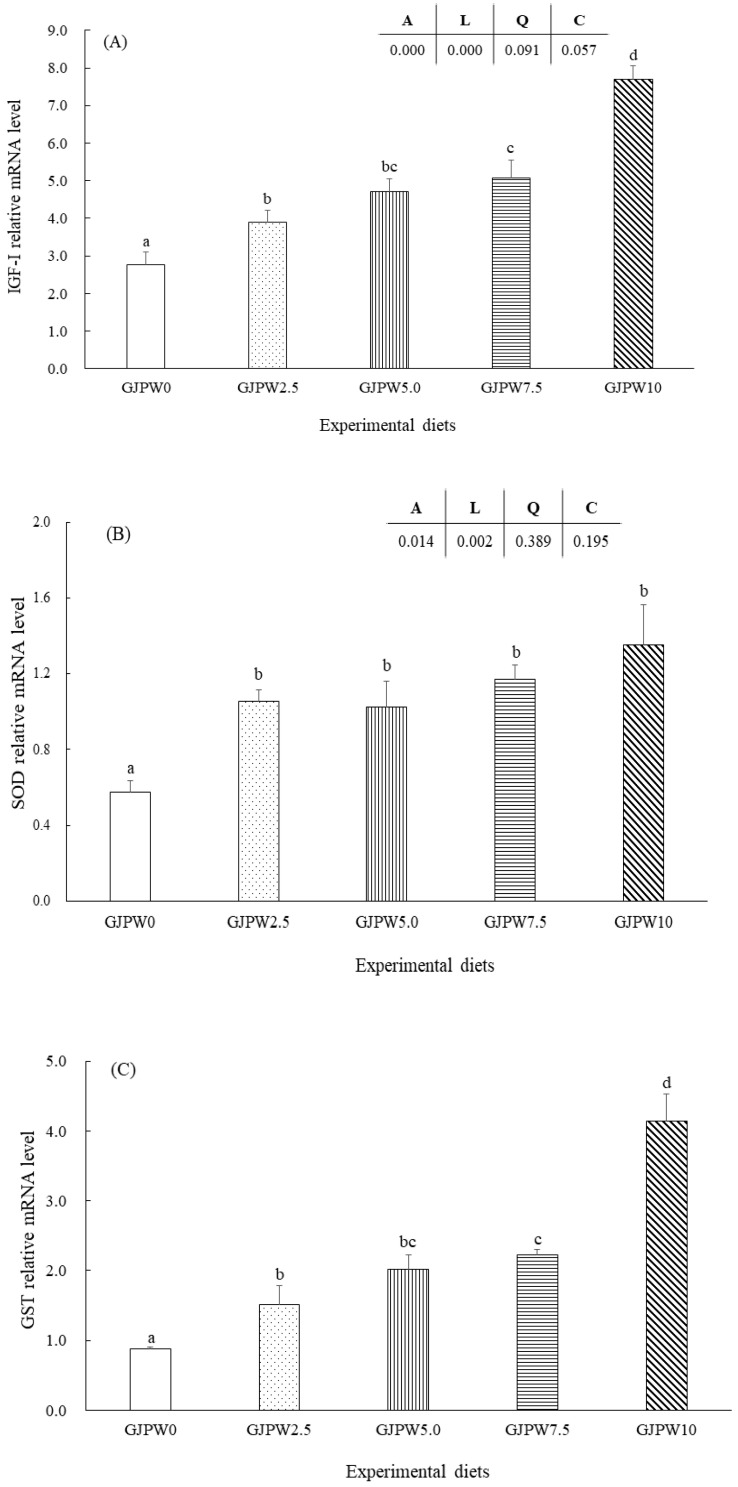
Relative expression [insulin-like growth factor (IGF-1) (**A**), superoxide dismutase (SOD) (**B**), glutathione S-transferase (GST) (**C**), and catalase (CAT) (**D**)] in liver samples of juvenile black rockfish fed with different levels of garlic juice processing waste (GJPW) for 8 weeks. All data are shown as mean ± SE of three replicates. Bars with different letters show statistically significant differences (*p* < 0.05).

**Figure 2 animals-12-03512-f002:**
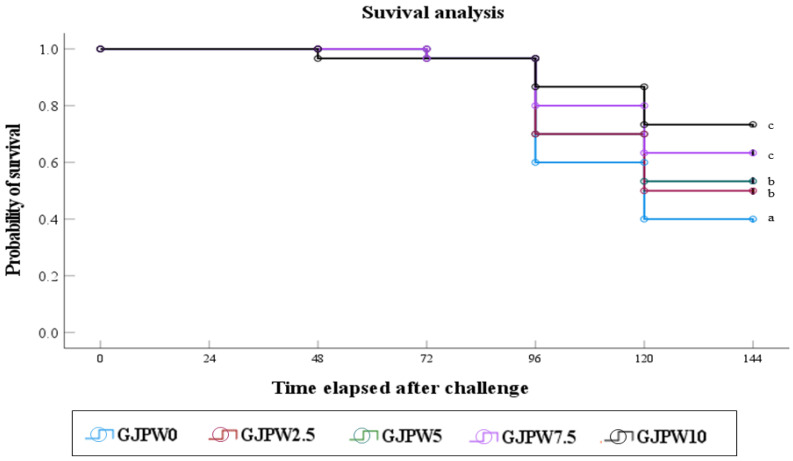
The survival of juvenile black rockfish fed experimental diets with different levels of inclusion of garlic juice processing waste (GJPW) for eight weeks and then infected by *Streptococcus iniae*. Values are means of triplicate groups. Different letters indicate significant differences (*p* < 0.001; log-rank and Wilcoxon tests). GJPW: garlic juice processing waste.

**Table 1 animals-12-03512-t001:** Total phenolic and flavonoid content and antioxidant activities of ethanol extract from garlic juice processing waste (GJPW).

	Chemical Compounds		Radical Scavenging Activities		
	Total Phenolics (mg/100 g)	Total Flavonoids (mg/100 g)	Concentration (µg/mL)	DPPH (%)	ABTS (%)
GJPW	27.3 ± 1.58	36.8 ± 1.75	2000	48.52 ± 1.37	73.23 ± 2.53
			1000	28.80 ± 1.09	51.00 ± 1.16
			500	13.59 ± 1.08	29.82 ± 3.89
			250	4.74 ± 0.15	17.57 ± 3.27
			IC_50_	2002.53 ± 59.63	1164.13 ± 69.72

DPPH: 1,1–diphenyl–2–picrylhydrazyl; ABTS: 2,2′–azinobis–(3–ethylbenzothiazoline–6–sulfonate); GJPW: garlic juice processing waste.

**Table 2 animals-12-03512-t002:** Experimental diet formulation (g kg^−1^, dry matter basis).

	Experimental Diets				
	GJPW0	GJPW2.5	GJPW5	GJPW7.5	GJPW10
Pollock meal	500	500	500	500	500
Fermented soybean meal	115	115	115	115	115
Wheat flour	270	267.5	265	262.5	260
Garlic juice processing waste (GJPW ^a^)	0	2.5	5	7.5	10
Fish oil	45	45	45	45	45
Soybean oil	45	45	45	45	45
Vitamin premix ^b^	10	10	10	10	10
Mineral premix ^c^	10	10	10	10	10
Choline chloride	5	5	5	5	5
*Proximate composition* (*g kg*^−1^)					
Dry matter	949	957	955	952	956
Crude protein	506	505	512	512	506
Crude lipid	140	137	143	142	139
Ash	89	89	93	89	89

^a^ GJPW (garlic juice processing waste) was supplied by the Youngjin Health Food store (Daegu, Korea). ^b^ Vitamin premix contained the following amounts, which were diluted in cellulose (g kg^−1^ mix): L-ascorbic acid, 121.2; DL-α-tocopheryl acetate, 18.8; thiamin hydrochloride, 2.7; riboflavin, 9.1; pyridoxine hydrochloride, 1.8; niacin, 36.4; Ca-D-pantothenate, 12.7; myo-inositol, 181.8; D-biotin, 0.27; folic acid, 0.68; p-aminobenzoic acid, 18.2; menadione, 1.8; retinyl acetate, 0.73; cholecalciferol, 0.003; cyanocobalamin, 0.003. ^c^ Mineral premix contained the following ingredients (g kg^−1^ mix): MgSO_4_·7H_2_O, 80.0; NaH_2_PO_4_·2H_2_O, 370.0; KCl, 130.0; ferric citrate, 40.0; ZnSO_4_·7H_2_O, 20.0; Ca-lactate, 356.5; CuCl, 0.2; AlCl_3_·6H_2_O, 0.15; KI, 0.15; Na_2_Se_2_O_3_, 0.01; MnSO_4_·H_2_O, 2.0; CoCl_2_·6H_2_O, 1.0.

**Table 3 animals-12-03512-t003:** Primer pair sequences of the genes used for quantitative real-time PCR in juvenile black rockfish (*Sebastes schlegelii*).

Gene	Primer Sequences; Forward/Reverse (5′ ⟶ 3′)	Target Tissue	Amplicon (bp)	Accession No.
IGF-1	F: ACACCCTCTCCCTACTGCTG R: CACACAAATTGGAGCGTGTC	Liver	109	AF481856.1
SOD	F: GGATCATGCCGGTCCTACTG R: GCCCAGTGAGAGTGAGCATC	Liver	119	AY771324.1
GST	F: AATGGAGCACAAGTCCCAAG R: GGCTGTCTGGGATCAGTTTG	Liver	158	AY771323.2
CAT	F: GCATGGTCGAGACCTTGAAT R: GCCTCGGCATTGTACTTGTT	Liver	82	KM401562.1
*β*-Actin	F: AGAGCTACGAGCTGCCTGAC R: AGGAAAGAGGGCTGGAAGAG	Liver	88	KF430616.1

GJPW: garlic juice processing waste; IGF-1: insulin-like growth factor-1; SOD, superoxide dismutase; CAT, catalase; GST, glutathione S-transferase.

**Table 4 animals-12-03512-t004:** Growth performance of juvenile rockfish fed with different levels of garlic juice processing waste (GJPW) inclusion in diets.

Items	Experimental Diets					SEM	Orthogonal Contrast			Regression		
	GJPW0	GJPW2.5	GJPW5	GJPW7.5	GJPW10		Linear	Quadratic	Cubic	Model	*p*-Value	Adj. R^2^
IBW (g)	2.2	2.2	2.2	2.2	2.2	0.09	0.628	0.682	0.165	NR	–	–
FBW (g)	9.7 ^a^	10.4 ^b^	10.6 ^b^	10.5 ^b^	10.6 ^b^	0.44	0.011	0.059	0.281	L	0.016	0.669
SR (%)	97.8	97.8	100.0	96.7	98.9	1.13	0.820	0.857	0.501	NR	–	–
WG (g/fish)	7.4 ^a^	8.2 ^b^	8.3 ^b^	8.3 ^b^	8.4 ^b^	0.45	0.016	0.081	0.309	L	0.018	0.662
SGR (%/day)	2.61 ^a^	2.76 ^b^	2.77 ^b^	2.77 ^b^	2.78 ^b^	0.20	0.018	0.088	0.285	L	0.020	0.649
CF	1.79	1.76	1.77	1.79	1.77	0.11	0.688	0.385	0.129	NR	–	–
VSI (%)	7.78	7.75	7.77	7.92	7.81	0.37	0.632	0.941	0.507	NR	–	–
HSI (%)	2.10	2.13	2.08	2.14	2.17	0.31	0.689	0.761	0.861	NR	–	–
FC (g/fish)	9.0	9.2	9.1	9.0	9.0	0.44	0.226	0.341	0.356	NR	–	–
FE	0.85 ^a^	0.91 ^b^	0.91 ^bc^	0.95 ^c^	0.95 ^c^	0.10	0.001	0.335	0.447	L	0.000	0.705
PER	1.64 ^a^	1.76 ^b^	1.78 ^b^	1.84 ^b^	1.88 ^b^	0.18	0.001	0.355	0.447	L	0.000	0.640
PR (%)	27.4 ^a^	29.8 ^ab^	30.2 ^ab^	30.5 ^b^	31.5 ^b^	0.88	0.008	0.601	0.460	L	0.003	0.600

Values (means of triplicate) in the same column sharing the different superscript letter are significantly different (*p* < 0.05). GJPW: garlic juice processing waste; SEM: pooled standard error of treatment means; Adj. R^2^: adjusted R square; IBW: initial body weight; FBW: final body weight; SR: survival; WG: weight gain; SGR: specific growth rate; CF: condition factor; VSI: viscerosomatic index; HSI: hematosomatic index; FC: feed consumption; FE: feed efficiency; PER: protein efficiency ratio; PR: protein retention; L: linear; NR: no relationship.

**Table 5 animals-12-03512-t005:** Proximate composition (%, wet weight basis) of whole-body of juvenile rockfish fed with different levels of garlic juice processing waste (GJPW) inclusion in diets.

Items	Experimental Diets					SEM	Orthogonal Contrast			Regression		
	GJPW0	GJPW2.5	GJPW5	GJPW7.5	GJPW10		Linear	Quadratic	Cubic	Model	*p*-Value	Adj. R^2^
Moisture (%)	71.6	71.7	71.6	71.9	71.7	0.46	0.594	0.940	0.536	NR	–	–
Crude protein (%)	16.9	17.0	17.0	16.8	16.9	0.33	0.478	0.651	0.335	NR	–	–
Crude lipid (%)	5.9	6.0	6.0	5.9	5.9	0.30	0.569	0.502	0.266	NR	–	–
Ash (%)	4.3	4.4	4.5	4.4	4.4	0.33	0.713	0.537	0.854	NR	–	–

GJPW: garlic juice processing waste; Adj. R^2^: adjusted R square; NR: no relationship.

**Table 6 animals-12-03512-t006:** Hematologic parameters of juvenile rockfish fed with different levels of inclusion of garlic juice processing waste (GJPW) in diets.

Items	Experimental Diets					SEM	Orthogonal Contrast			Regression		
	GJPW0	GJPW2.5	GJPW5	GJPW7.5	GJPW10		Linear	Quadratic	Cubic	Model	*p*-Value	Adj. R^2^
AST (U/L)	153.0 ^b^	95.7 ^a^	93.0 ^a^	91.3 ^a^	91.0 ^a^	3.21	0.022	0.068	0.287	L	0.028	0.519
ALT (U/L)	34.7 ^b^	29.0 ^a^	29.0 ^a^	27.7 ^a^	26.7 ^a^	1.67	0.049	0.079	0.575	L	0.041	0.444
T-CHO (mg/dL)	209.3	193.7	181.3	178.7	178.7	3.88	0.230	0.574	0.991	NR	–	–
TP (g/dL)	4.4	5.3	4.9	5.0	5.6	0.53	0.146	0.933	0.114	NR	–	–
GLU (mg/dL)	63.0 ^b^	46.7 ^a^	40.7 ^a^	39.0 ^a^	37.3 ^a^	1.98	0.002	0.081	0.497	L	0.002	0.532
Lysozyme (U/mL)	57.0 ^a^	60.5 ^b^	60.7 ^b^	60.9 ^b^	64.2 ^b^	1.07	0.002	0.929	0.105	L	0.002	0.550
SOD (U/mL)	5.3 ^a^	6.7 ^b^	6.9 ^b^	7.0 ^b^	7.0 ^b^	0.58	0.007	0.059	0.305	L	0.010	0.515
CAT (nmol/min/mL)	2.1 ^a^	2.5 ^b^	2.6 ^b^	2.6 ^b^	2.7 ^b^	0.37	0.010	0.147	0.323	L	0.008	0.526
GSH (µM)	15.2 ^a^	17.4 ^b^	17.7 ^b^	17.4 ^b^	17.4 ^b^	0.76	0.044	0.046	0.277	L	0.042	0.493

Values are means from triplicated groups of fish where the values in the same column sharing the different superscript letter are significantly different (*p* < 0.05). GJPW: garlic juice processing waste; SEM: pooled standard error of treatment means; Adj. R^2^: adjusted R square; AST: aspartate aminotransferase; ALT: alanine aminotransferase; T-CHO: total cholesterol; TP: total protein; GLU: glucose; SOD: superoxide dismutase; CAT: catalase, GSH: glutathione; L: linear; NR: no relationship.

**Table 7 animals-12-03512-t007:** Digestive enzyme activities of juvenile rockfish fed with different levels of garlic juice processing waste (GJPW) inclusion in diets.

Items	Experimental diets					SEM	Orthogonal contrast			Regression		
	GJPW0	GJPW2.5	GJPW5	GJPW7.5	GJPW10		Linear	Quadratic	Cubic	Model	*p*-Value	Adj. R^2^
Amylase (U/L)	46.5 ^a^	54.0 ^b^	60.0 ^b^	64.4 ^b^	61.0 ^b^	1.68	0.000	0.003	0.180	L	0.003	0.615
Trypsin (U/L)	7.5 ^a^	9.5 ^b^	11.2 ^c^	11.7 ^c^	13.5 ^d^	0.63	0.000	0.354	0.216	L	0.000	0.902
Lipase (U/L)	1.3 ^a^	2.4 ^b^	2.4 ^b^	2.8 ^b^	2.7 ^b^	0.36	0.003	0.082	0.541	L	0.001	0.701

Values are means of triplicated groups of fish, where the values in the same column sharing the different superscript letter are significantly different (*p* < 0.05). GJPW: garlic juice processing waste; SEM: pooled standard error of treatment means; Adj. R^2^: adjusted R square; L: linear.

## Data Availability

Data available on reasonable request.

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
