# Peer review of "Evaluation of Garlic Juice Processing Waste Supplementation in Juvenile Black Rockfish (Sebastes schlegelii) Diets on Growth Performance, Antioxidant and Digestive Enzyme Activity, Growth- and Antioxidant-Related Gene Expression, and Disease Resistance against Streptococcus iniae"

_animals, 2022, doi:10.3390/ani12243512_

Round 1
Reviewer 1 Report
Evaluation of Garlic Juice Processing Waste Supplementation in Juvenile Black Rockfish (Sebastes schlegelii) Diets on Growth Performance, Antioxidant and Digestive Enzyme Activity, Growth- and Antioxidant-related Gene Expression, and Disease Resistance against Streptococcus iniae
General Comments:
The work presented here deals with garlic by-products and their effect on the growth and health of the Black rockfish Sebastes schlegelii. There are countless works about phytobiotics. In this case, they typified the by-product by measuring the phenolic and flavonoid compounds to determine the effect of such by-product. However, it is crucial to quantify the total amount of phenolic compounds and flavonoids in the feed to know precisely the amount present.
How the authors explain that with so high difference in the IGF1 expression, growth fail to show so high disparity.
Particular comments
It is essential to give details of the techniques used in the kits for enzyme activity and antioxidant enzymes.
If garlic juice has an antioxidant action, how do the authors explain why the antioxidant enzymes were expressed more in the experimental diets than in the control group?
The effect on survival after the challenge is apparent, although there were no differences in growth. Could the energy expended for the expression of the endogenous enzymes not have affected the protein-sparing effect?
If phenolic compounds and flavonoids are antioxidants, why does the activity of endogenous antioxidant enzymes increase? Shouldn't it be the other way around?
Line 392, why didn't they evaluate the digestibility?
Line 432, please use among when comparing more than two variables. The word between is used only with two.
Line 467, same as previously indicated. If the use of antioxidants in the diet is to neutralize the ROS produced and thus avoid the energy expenditure of organisms when producing their endogenous enzymes. How do the authors explain this?
Author Response
We are sincerely grateful for your thorough consideration and scrutiny of our manuscript, “Evaluation of garlic juice processing waste supplementation in juvenile black rockfish (Sebastes schlegelii) diets on growth performance, antioxidant and digestive enzyme activity, growth- and antioxidant-related gene expression, and disease resistance against Streptococcus iniae”. Through the accurate comments made by the reviewers, we better understand the critical issues in this paper. We have revised the manuscript according to the Reviewer’s suggestions. We hope that our revised manuscript will be considered and accepted for publication in Animals. We acknowledge that the scientific and clinical quality of our manuscript was improved by the scrutinizing efforts of the reviewers and editors. Point-by-point responses to the reviewers’ comments are provided below.
Point 1: The work presented here deals with garlic by-products and their effect on the growth and health of the Black rockfish Sebastes schlegelii. There are countless works about phytobiotics. In this case, they typified the by-product by measuring the phenolic and flavonoid compounds to determine the effect of such by-product. However, it is crucial to quantify the total amount of phenolic compounds and flavonoids in the feed to know precisely the amount present.
Response 1: Thank you for reviewer’s careful point. According to the reviewer's opinion, it would be good to analyse the total amount of phenolic compounds and flavonoids in the experimental diets. However, in this study, the analysis of phenolic compounds and flavonoids in the experimental diets was not conducted. However, the content of phenolic compounds and flavonoids in the ginger juice by-products was analysed, and the inclusion level of juice by-products was increased in the preparation of this experimental diets, and accordingly, the content of phenolic compounds and flavonoids in the experimental diets was also increased. This is supported by the results of feeding trial results (growth performance, digestvie enzme, antioxidant capacity, and related gene expression).
Point 2: How the authors explain that with so high difference in the IGF1 expression, growth fail to show so high disparity.
Response 2: Thank you for reviewer’s careful point. However, there is something vague about the reviewer's opinion. In this stuy, we have already argued in the discussing of this manuscirpt that the higher growth performance was recorded by fish fed diet supplemented with GJPW compared to the control (GJPW0) treatment, also the relative expression of IGF-1 gene mRNA exhibited the same trend. This finding indicated that the supplementation of 2.5 to 10 g of GJPW in diet can have a positive regulatory effect on the transcription of the IGF-1, which can eventually involve hyperplastic and hypertrophic muscular growth.
Point 3: It is essential to give details of the techniques used in the kits for enzyme activity and antioxidant enzymes.
Response 3: Thank you for reviewer’s careful point. We add the details of method in assay kit.
Point 4: If garlic juice has an antioxidant action, how do the authors explain why the antioxidant enzymes were expressed more in the experimental diets than in the control group?
Response 4: In many studies, antioxidants in feed additives have been reported to be very effective in improving the antioxidant ability of organisms. Especially, tt is well known that Nrf2-Keap1 signaling pathway plays a vital role in antioxidant system (Yang et al., 2021). Nrf2 could dissociate from Keap1 and further regulate the transcription levels of downstream antioxidant genes in nucleus under stressed conditions after activation (Bellezza et al., 2018). Substantial evidences in mammal have demonstrated that polyphenols and flavonoids could improve the antioxidant capacity by activating Nrf2 (Hussain et al., 2016; Sharma et al., 2016).
In addition, as discussed in this manuscript, the addition of garlic as a feed additive had a significant effect on the improvement of antioxidant enzyme activity in aquatic animals.
Yang, G., Yu, R., Geng, S., Xiong, L., Yan, Q., Kumar, V., ... & Peng, M. (2021). Apple polyphenols modulates the antioxidant defense response and attenuates inflammatory response concurrent with hepatoprotective effect on grass carp (Ctenopharyngodon idellus) fed low fish meal diet. Aquaculture, 534, 736284.
Bellezza, I., Giambanco, I., Minelli, A., & Donato, R. (2018). Nrf2-Keap1 signaling in oxidative and reductive stress. Biochimica et Biophysica Acta (BBA)-Molecular Cell Research, 1865(5), 721-733.
Hussain, T., Tan, B., Yin, Y., Blachier, F., Tossou, M. C., & Rahu, N. (2016). Oxidative stress and inflammation: what polyphenols can do for us?. Oxidative medicine and cellular longevity, 2016.
Sharma, S., Rana, S., Patial, V., Gupta, M., Bhushan, S., & Padwad, Y. S. (2016). Antioxidant and hepatoprotective effect of polyphenols from apple pomace extract via apoptosis inhibition and Nrf2 activation in mice. Human & Experimental Toxicology, 35(12), 1264-1275.
Point 5: The effect on survival after the challenge is apparent, although there were no differences in growth. Could the energy expended for the expression of the endogenous enzymes not have affected the protein-sparing effect?
Response 5: Thank you for reviewer’s careful point. We respect the reviewer's opinion. However, in this study, both the crude protein and the crude content of the experimental diets were similarly prepared and prepared. Therefore, it seems unreasonable to discuss the protein-sparing effect of the supplementation garlic juice processing waste. However, research on energy budget according to phyto-additives such as garlic juice processing waste needs to be conducted in the future.
Point 6: If phenolic compounds and flavonoids are antioxidants, why does the activity of endogenous antioxidant enzymes increase? Shouldn't it be the other way around?
Response 6: This is a point similar to point 4 and is the same as response 4.
Point 7: Line 392, why didn't they evaluate the digestibility?
Response 7: This study focuses on the effect of garlic juice processing waste as feed additives on digestive enzymes, antioxidant capacity, and disease resistance of juvenile rockfish. If the digestibility experiment was conducted according to the reviewer's opinion, it is thought that the scientific basis of the results of this study will be further supported. However, in this study, feed utilization is increased and growth is enhanced by improving intestinal digestive enzyme activity by garlic juice processing waste, so garlic juice processing waste is considered to be very effective in digestive utilization. However, in the future, digestibility evaluation will be conducted in a study on the efficacy of garlic juice processing waste additives in low-fish meal diet.
Point 8: Line 432, please use among when comparing more than two variables. The word between is used only with two.
Response 8: Thank you for reviewer’s careful point. We rewirte.
Point 9: Line 467, same as previously indicated. If the use of antioxidants in the diet is to neutralize the ROS produced and thus avoid the energy expenditure of organisms when producing their endogenous enzymes. How do the authors explain this?
Response 9: Similar to the previous response, we sympathize very much with the reviewer's opinion. However, because this study does not focus on energy budget such as antioxidant production energy expenditure according to garlic juice processing waste, it is very limited to consider the discussion related to energy expenditure. However, we think it is necessary to study the clear mechanism for this in the future.
Reviewer 2 Report
In this study, the recommended dosage of Garlic Juice Processing Waste was evaluated by growth performance, enzyme activity, quantitative PCR and Challenge test. However, the calculation of growth data is wrong and should be double checked.
In addition, there are some problems in the manuscript, such as the abbreviations are inconsistent and the calculation formulas and table data are not uniform, etc.
1. Introduction
Line 49-51: please rewrite this sentence.
Line 71-73: Please provide the source of the data.
Line 76-77: Please add the references.
2. Materials and Methods
Line 118: A fish with an initial weight of only 2.2g is fed a feed with a particle size of 3mm? Would it affect the feed intake and growth of fish?
Line 122-123: Please add brackets.
Line 128-129: The tank seems to be too small.
Line 130: Please delete “h”.
Line157-164: Was there a pre-experiment done to determine the semi-lethal concentration?
Line 193: Are the experimental diet and whole-body samples both dried at 105°C for 24h?
Line 222: Is the abbreviation for insulin-like growth factor "IGF-1" or "IGF-I"?
Line 247: Are the calculations of WG, FE, and PER correct?
3. Results
Line 283-285: Please note the grammar.
Line 308-310: Please check the result.
Line 327: The same abbreviation in the main text only needs to be added once.
Line 325-327: Please supplement “p<0.05”.
Line 335-341: Please supplement “p<0.05” or “p>0.05”.
4. Discussion
Line 387: Please delete “in”.
Line 414: Please delete “a”.
Line 432: Please change “difference” to “differences”.
Line 436: Please change “is” to “are”.
Line 436: Please change “indicator” to “indicators”.
Line 446: Please rewrite this sentence.
The discussion should be revised, focused on the main results of the manuscript, and enhance the connections between paragraphs
4. Table and figure
Figure 2: Color scheme is not clear, please change the color.
Author Response
We are sincerely grateful for your thorough consideration and scrutiny of our manuscript, “Evaluation of garlic juice processing waste supplementation in juvenile black rockfish (Sebastes schlegelii) diets on growth performance, antioxidant and digestive enzyme activity, growth- and antioxidant-related gene expression, and disease resistance against Streptococcus iniae”. Through the accurate comments made by the reviewers, we better understand the critical issues in this paper. We have revised the manuscript according to the Reviewer’s suggestions. We hope that our revised manuscript will be considered and accepted for publication in Animals. We acknowledge that the scientific and clinical quality of our manuscript was improved by the scrutinizing efforts of the reviewers and editors. Point-by-point responses to the reviewers’ comments are provided below.
Point 1: In this study, the recommended dosage of Garlic Juice Processing Waste was evaluated by growth performance, enzyme activity, quantitative PCR and Challenge test. However, the calculation of growth data is wrong and should be double checked.
Response 1: Thank you for reviewer’s careful point. We found an error in our growth formula and corrected it.
Point 2: In addition, there are some problems in the manuscript, such as the abbreviations are inconsistent and the calculation formulas and table data are not uniform, etc.
Response 2: Thank you for reviewer’s careful point. As mentioned by the reviewer, the problem was corrected by thorough double-checking.
Point 3: 1. Introduction
Line 49-51: please rewrite this sentence.
Response 3: Thank you for reviewer’s careful point. We rewrite the sentence.
Point 4: Line 71-73: Please provide the source of the data.
Response 4: Thank you for reviewer’s careful point. We provide the source of the data.
Point 5: Line 76-77: Please add the references.
Response 5: Thank you for reviewer’s careful point. We add the references.
Point 6: 2. Materials and Methods
Line 118: A fish with an initial weight of only 2.2g is fed a feed with a particle size of 3mm? Would it affect the feed intake and growth of fish?
Response 6: : Thank you for reviewer’s careful point. The size of the commercial formulated feed for 2-3 g juvenile rockfish in Korea was 2-3 mm, and this experimental diets also had no effect on the feeding results according to the feed size.
Point 8: Line 122-123: Please add brackets.
Response 8: Thank you for reviewer’s careful point. We add the brackets.
Point 9: Line 128-129: The tank seems to be too small.
Response 9: Thank you for reviewer’s careful point. Many studies have already been conducted in a tank similar to the size of the tank used in the experiment (Bai and Lee, 1998; Yoo and Bai et al., 2014; Jeon et al., 2014), and in particular, considering the size of the juvenile rockfish used in this experiment, the size of the experimental tank and density did not significantly affect the results of feeding trial.
Bai, S. C., & Lee, K. J. (1998). Different levels of dietary DL-α-tocopheryl acetate affect the vitamin E status of juvenile Korean rockfish, Sebastes schlegeli. Aquaculture, 161(1-4), 405-414.
Yoo, G., & Bai, S. C. (2014). Effects of the dietary microbial phytase supplementation on bioavailability of phosphorus in juvenile olive flounder Paralichthys olivaceus fed soybean meal based diets. Fisheries and aquatic sciences, 17(3), 319-324.
Jeon, G. H., Kim, H. S., Myung, S. H., & Cho, S. H. (2014). The effect of the dietary substitution of fishmeal with tuna by‐product meal on growth, body composition, plasma chemistry and amino acid profiles of juvenile K orean rockfish (S ebastes schlegeli). Aquaculture Nutrition, 20(6), 753-761.
Point 10: Line 130: Please delete “h”.
Response 10: Thank you for reviewer’s careful point. We delete “h”.
Point 11: Line157-164: Was there a pre-experiment done to determine the semi-lethal concentration?
Response 11: Thank you for reviewer’s careful point. Naturally, after conducting a preliminary semi-lethal concentration experiment, the range of artificial infection concentrations was selected.
Point 12: Line 193: Are the experimental diet and whole-body samples both dried at 105°C for 24h?
Response 12: Thank you for reviewer’s careful point. The experimental diet and whole-body samples both wer dried at 105°C for 24h to measure the moisture content accroding to AOAC.
Point 13: Line 222: Is the abbreviation for insulin-like growth factor "IGF-1" or "IGF-I"?
Response 13: Thank you for reviewer’s careful point. This means IGF-1 and has all been revised.
Point 14: Line 247: Are the calculations of WG, FE, and PER correct?
Response 14: Thank you for reviewer’s careful point. We found an error in our growth formula and corrected it.
Point 15: 3. Results
Line 283-285: Please note the grammar.
Response 15: Thank you for reviewer’s careful point. We check the grammar and revised.
Point 16: Line 308-310: Please check the result.
Response 16: We check the result and revised.
Point 17: Line 327: The same abbreviation in the main text only needs to be added once.
Response 17: We check the result and revised.
Point 18: Line 325-327: Please supplement “p<0.05”.
Response 18: Thank you for reviewer’s careful point. We add“p<0.05”.
Point 19: Line 335-341: Please supplement “p<0.05” or “p>0.05”.
Response 19: Thank you for reviewer’s careful point. We add“p<0.05”.
Point 20: 4. Discussion
Line 387: Please delete “in”.
Response 20: Thank you for reviewer’s careful point. We delete it.
Point 21: Line 414: Please delete “a”.
Response 21: Thank you for reviewer’s careful point. We delete it.
Point 22: Line 432: Please change “difference” to “differences”.
Response 22: Thank you for reviewer’s careful point. We change it.
Point 23: Line 436: Please change “is” to “are”.
Response 23: Thank you for reviewer’s careful point. But, since the subject indicates the activity, it is considered correct to use singular verbs.
Point 24: Line 436: Please change “indicator” to “indicators”.
Response 24: Thank you for reviewer’s careful point. We change it.
Point 25 Line 446: Please rewrite this sentence.
Response 25: Thank you for reviewer’s careful point. We rewirte sentence.
Point 26: The discussion should be revised, focused on the main results of the manuscript, and enhance the connections between paragraphs
Response 26: Thank you for reviewer’s careful point. We think the discursion has improved by applying all the careful points of the reviewer, and once again, thank you for the careful review.
Point 27: 4. Table and figure
Figure 2: Color scheme is not clear, please change the color.
Response 27: Thank you for reviewer’s careful point. We change the color.
Round 2
Reviewer 1 Report
General Comments:
This present manuscript is a straightforward experiment. Even if I agree with the author's comments on the first review, I cannot entirely agree with the explanation of why the product was not fully characterized. Plenty of works have been described with alisine from garlic; however, since this product is a by-product of the given process, it would be essential to do the characterization. Furthermore, there are similar studies; therefore, it is crucial to be more specific to be unique in this work.
However, the terminology in this field is often misinterpreted. In this case, the impression that garlic juice has antioxidant activity is incorrect. However, it could be that it stimulates the proliferation of antioxidant enzymes. Therefore, it is crucial to be more precise and discuss possible mechanisms. I recommend reading Firmino et al., 2021. If differences between the igf1 and growth are seen, it could be possible that SOD upregulation, as well as the rest of the enzymes, could use the energy instead of growth? Does this mean that garlic is a stimulator of antioxidant enzymes and not an antioxidant per se? as assumed in line 368?
Moreover, it may be a mistake, but the authors wrote Ginger in the answers. Was Ginger also used? Or is it simply a writing error?
The following comment is difficult to follow: "However, the content of phenolic compounds and flavonoids in the ginger juice by-products was analyzed, and the inclusion level of juice by-products was increased in the preparation of this experimental diets, and accordingly, the content of phenolic compounds and flavonoids in the experimental diets was also increased." It was increased, so this is obvious as part of the experimental design, correct?, but it has nothing to do with the analysis, which is essential.
Could the author's correct Table 4? The SGR from the GJW10 treatment has a similar weight gain as GJW5 (from 2.2 to 10.6), which does not agree with the gain per gram being 8.3 vs. 8.4. Resulting in a different SGR. Could authors verify that information?
Particular comments
Line 434 Why not hepatic histology it was performed to prove this fact?
Line 443 this comment here is speculation.
Line 452 to 460, please do not confound antioxidant activity vs. stimulation of antioxidant enzymes.
Author Response
Point 1: This present manuscript is a straightforward experiment. Even if I agree with the author's comments on the first review, I cannot entirely agree with the explanation of why the product was not fully characterized. Plenty of works have been described with alisine from garlic; however, since this product is a by-product of the given process, it would be essential to do the characterization. Furthermore, there are similar studies; therefore, it is crucial to be more specific to be unique in this work.
Response 1: Thank you again for the reviewer's meticulous review. This study is not a simple experiment. As the reviewer said, it is thought that a good scientific basis would be supported if the characteristics of by-products were tested through more analysis methods. However, the purpose of this study is to analyze the antioxidant content of garlic juice by-products evaluated as potential feed additives in Lee et al. (2021)’s study, and to evaluate growth, antioxidant and digestive enzyme activity and related gene expression levels after 8 weeks of feeding experiments. For commercial or field application of by-products, additional analysis of various functional substances of the reviewer's mentioned by-products must be made. Unfortunately, however, we would appreciate it if you could understand that this study did not focus on analyzing various functional substances, but for the purpose mentioned above.
Point 2: However, the terminology in this field is often misinterpreted. In this case, the impression that garlic juice has antioxidant activity is incorrect. However, it could be that it stimulates the proliferation of antioxidant enzymes. Therefore, it is crucial to be more precise and discuss possible mechanisms. I recommend reading Firmino et al., 2021. If differences between the igf1 and growth are seen, it could be possible that SOD upregulation, as well as the rest of the enzymes, could use the energy instead of growth? Does this mean that garlic is a stimulator of antioxidant enzymes and not an antioxidant per se? as assumed in line 368?
Response 2: Thank you for reviewer’s careful point. We even appreciate the reviewer's recommendation of the article (Firmino et al., 2021). However, we hope that the paper we reviewed is correct. It is true that there are very limited parts to fully understand and grasp the mechanism of the relationship between the accurate antioxidant function and the growth function of garlic juice by-products through this study. However, as carried out in this study, the antioxidant activity (DPPH and ABST) of garlic juice by-product was analyzed to prove its activity, and when added to the feed, the growth capacity and the activity of antioxidant enzymes in plasma were also demonstrated due to increased the expression of antioxidant enzymes and growth-related genes. This is thought to be an original part of this study, but it is thought that a study of the functional mechanism of phytogenic growth and immunity in fish should be conducted in the future as the reviewer's opinion.
Firmino, J. P., Galindo-Villegas, J., Reyes-López, F. E., & Gisbert, E. (2021). Phytogenic bioactive compounds shape fish mucosal immunity. Frontiers in immunology, 12, 695973.
Point 3: Moreover, it may be a mistake, but the authors wrote Ginger in the answers. Was Ginger also used? Or is it simply a writing error?
Response 3: Thank you for reviewer’s careful point. It's my mistake. Garlic is right, not ginger.
Point 4: The following comment is difficult to follow: "However, the content of phenolic compounds and flavonoids in the ginger juice by-products was analyzed, and the inclusion level of juice by-products was increased in the preparation of this experimental diets, and accordingly, the content of phenolic compounds and flavonoids in the experimental diets was also increased." It was increased, so this is obvious as part of the experimental design, correct?, but it has nothing to do with the analysis, which is essential.
Response 4: Thank you for reviewer’s careful point. We misunderstood the reviewer's opinion a lot. However, it was accepted as an opinion that it was necessary to analyze the antioxidant content of the experimental diets. Please understand generously that the analysis of the antioxidant content of the experimental feed was not set in the purpose and hypothesis of this study.
Point 5: Could the author's correct Table 4? The SGR from the GJW10 treatment has a similar weight gain as GJW5 (from 2.2 to 10.6), which does not agree with the gain per gram being 8.3 vs. 8.4. Resulting in a different SGR. Could authors verify that information?
Response 5: Thank you for the reviewer’s careful point. We are sorry that we couldn't double-check even though the reviewer mentioned it. The formula for weight gain was modified, and the results of Table 4 were confirmed.
Point 6: Line 434 Why not hepatic histology it was performed to prove this fact?
Response 6: Thank you for the reviewer’s careful point. In the reviewer's opinion, it is regrettable that scientific proof of the results through histological analysis was not made, but as mentioned above, liver histological analysis was not included for the purpose of this study. Also, unfortunately, please understand that further histological analysis preparation (hepatic tissue preparation) cannot be made.
Point 7: Line 443 this comment here is speculation.
Response 7: Thank you for the reviewer’s careful point. We agree with your opinion and exclude the speculation in this sentence.
Point 8: Line 452 to 460, please do not confound antioxidant activity vs. stimulation of antioxidant enzymes.
Response 8: Thank you for the reviewer’s careful point. It is hard to find the part where the meaning is cofounded in the part you pointed out. However, all authors respect the reviewer's opinion.
Reviewer 2 Report
It could be accepted after the reversion.
Author Response
Thank you again for your review so far.